# Generation of Hematopoietic-Like Stem Cells from Adult Human Peripheral Blood Following Treatment with Platelet-Derived Mitochondria

**DOI:** 10.3390/ijms21124249

**Published:** 2020-06-15

**Authors:** Haibo Yu, Wei Hu, Xiang Song, Dante Descalzi-Montoya, Zheng Yang, Robert Korngold, Yong Zhao

**Affiliations:** Center for Discovery and Innovation, Hackensack Meridian Health, Nutley, NJ 07110, USA; Haibo.Yu@HMH-CDI.org (H.Y.); whu2@stevens.edu (W.H.); Xiang.Song@HMH-CDI.org (X.S.); DDescalz@kean.edu (D.D.-M.); Zheng.Yang@HMH-CDI.org (Z.Y.); Robert.Korngold@HMH-CDI.org (R.K.)

**Keywords:** mitochondria, PB-IPC, hematopoietic stem cells, differentiation, blood

## Abstract

Adult stem cells represent a potential source for cellular therapy to treat serious human diseases. We characterized the insulin-producing cells from adult peripheral blood (designated PB-IPC), which displayed a unique phenotype. Mitochondria are normally located in the cellular cytoplasm, where they generate ATP to power the cell’s functions. Ex vivo and in vivo functional studies established that treatment with platelet-derived mitochondria can reprogram the transformation of adult PB-IPC into functional CD34^+^ hematopoietic stem cells (HSC)-like cells, leading to the production of blood cells such as T cells, B cells, monocytes/macrophages, granulocytes, red blood cells, and megakaryocytes (MKs)/platelets. These findings revealed a novel function of mitochondria in directly contributing to cellular reprogramming, thus overcoming the limitations and safety concerns of using conventional technologies to reprogram embryonic stem (ES) and induced pluripotent stem (iPS) cells in regenerative medicine.

## 1. Introduction

Stem-cell research has the potential to revolutionize treatments for certain life-changing injuries and devastating human diseases such as diabetes and Alzheimer’s disease. To date, researchers have characterized multiple types of human stem cells with varying levels of potential for regeneration. Animal studies and human trials have demonstrated stem cells’ translational capability to treat human diseases. For over three decades now, the most common stem-cell therapy approved by the FDA has been hematopoietic cell transplantation (HCT) (also termed hematopoietic stem cell transplantation, HSCT) for the treatment of bone marrow failure, malignant blood disorders, post-chemotherapy and/or -radiation cell regeneration, genetically based blood disorders, and autoimmune diseases [1,2,3]. However, several major limitations have restricted the broad clinical application of allogeneic HCT, including the difficulty in identifying a fully human leukocyte antigen (HLA)-matched or haploidentical donor, the scarcity of CD34^+^ hematopoietic stem cells (HSCs) amongst all sources of harvested cells (≤1%) [4], and, in particular, the incidence of graft-versus-host disease (GVHD), opportunistic infections, relapse of primary disease, and toxicities associated with immunosuppressive drugs and radiation. An autologous source of HSC would address the problems of matching and GVHD, but engraftment could still be hampered by the limited number of CD34^+^ HSCs. Since the success rate of engraftment for clinical HCT is correlated with the number of functional CD34^+^ hematopoietic progenitor cells (HPCs) and HSCs in the transplant [5,6], researchers have evaluated whether embryonic stem (ES) cells and induced pluripotent stem (iPS) cells can be manipulated to produce HSCs through reprogramming by small molecules or by viral transduction of transcription factors [7,8,9,10,11,12]. Thus far, these approaches have been limited by an inability to generate true functional HSCs in sufficient numbers for therapeutic use, as well as safety and ethical concerns and potential immune rejection issues to ES or iPS derivatives [5,13]. Alternative approaches are needed to circumvent these limitations.

We previously characterized a new type of multipotent stem cell from human cord blood (designated cord-blood-derived multipotent stem cells, CB-SCs) [14] that is distinguished from other known stem-cell types, including HSCs and mesenchymal stem cells (MSCs). Based on the comprehensive immune modulation characteristics of CB-SCs [15], we developed the stem cell educator (SCE) therapy, which consists of isolating the patient’s blood cells through a blood cell separator, co-culturing the patient’s immune cells with adherent CB-SCs in vitro, and returning the “educated” immune cells to the patient’s circulation [15,16,17]. Our clinical trials indicated that SCE therapy reverses multiple immune dysfunctions and corrects autoimmune memory [17], promotes regeneration of islet β cells, improves metabolic control for the treatment of type 1 diabetes (T1D) [15,17,18] and type 2 diabetes (T2D) [16,18], and can be used to treat other autoimmune diseases such as for hair regrowth in alopecia areata (AA) [19]. Using a similar approach previously utilized for an isolation of CB-SC [15,16,17,18,19], we characterized adult peripheral blood insulin-producing cells (PB-IPC) [20] from adult peripheral blood by virtue of their ability to adhere to hydrophobic surfaces with positive charge. While investigating the mechanisms underlying the effects of SCE therapy in ex vivo studies [18], we observed the migration of platelet-released mitochondria to pancreatic islets, where they can enhance islet-β-cell function and C-peptide (an insulin byproduct) production, supporting clinical observations of the long-lasting improved health status in subjects with T1D and T2D after receiving SCE therapy [18]. Subsequently, we sought to determine whether platelet-derived mitochondria could directly influence the functions or reprogram the differentiation of adult PB-IPCs. Notably, our recent ex vivo studies established that mitochondrion-induced PB-IPCs (miPB-IPCs) can give rise to retinal pigment epithelium (RPE) cells and neuronal cells in the presence of different inducers [21]. Here, we found that platelet-derived mitochondria may be used to transform PB-IPCs into CD34^+^-hematopoietic-stem-cell-like cells, giving rise to multiple blood cell lineages.

## 2. Results

### 2.1. Ex Vivo Differentiation of Mitochondrion-Induced PB-IPCs (miPB-IPCs) into the Mitochondrion-Induced CD34^+^-HSC-Like Cells (miCD34^+^ HSCs) after Treatment with Platelet-Derived Mitochondria

Platelets are enucleate cells without human genomic DNA. We obtained high-purity apheresis platelets (>99% CD41^+^CD42^+^ platelets [18]) from the New York Blood Center for our studies. To determine the purity of the mitochondria isolated from platelets, different markers were applied by flow cytometry including MitoTrack Deep Red staining, anti-cytochrome C and anti-heat shock protein (HSP) 60 Abs for mitochondrial markers, calnexin for endoplasmic reticulum (ER), and GM130 for Golgi apparatus. Flow cytometry demonstrated that 99% of isolated mitochondria were positive for MitoTrack Deep Red, HSP 60, and cytochrome C; there were about 5% cytochrome C^+^ calnexin^+^ cells and 4% cytochrome C^+^GM130^+^ (Figure 1A). These double-positive staining results may have been caused by the interaction and conjugation of mitochondria with the ER or Golgi apparatus, respectively. Flow cytometry analysis demonstrated that the purity of the isolated mitochondria was ≥90%. (Figure 1A). We prepared the purified platelet-derived mitochondria from autologous or allogeneic peripheral blood, as described previously [18], and treated PB-IPCs that were isolated and expanded from blood samples of adult donors at the New York Blood Center (*n* = 51; mean age of 48.76 ± 14.97; age range from 18 to 72 years old; 24 males and 27 females). Notably, the expression of the HSC marker CD34 was upregulated in PB-IPCs after the treatment with mitochondria. A phenotypic analysis of miPB-IPCs after two weeks of mitochondrial treatment was striking in that the expression of CD34 on miPB-IPCs increased from 0.71% ± 0.25% to 14.8% ± 3.1% (*p* = 7.88 × 10^6^, *n* = 5) (Figure 1B). Using an optimized panel of cell markers [22], we found that mitochondrion-induced CD34^+^ (miCD34^+^) cells displayed a phenotype of CD34^+^CD38^−/low^CD45RA^−^CD49f^+^CD90^+^Flt3^−/low^CD7^+^CD10^+^CD71^+^BAH1^−/low^ (14.8% ± 3.1%, *n* = 5) (Figure 1C). In comparison to regular blood CD34^+^CD45RA^−^CD90^+^Flt3^−/low^CD7^+^CD71^+^ HSCs (0.49% ± 0.19%, *n* = 4) from non-mobilized healthy donors, the miCD34^+^ cells expressed similar surface markers as CD34^+^CD45RA^−^CD90^+^Flt3^−/low^CD7^+^CD71^+^ (15.3% ± 2.9%, *n* = 5, *p* < 0.01), but higher levels of CD10 (a marker defining human lymphoid progenitors [23]) (99.4% ± 0.36% versus 20.6% ± 3.1%, *p* < 0.01), CD49f (a common biomarker for most populations of stem cells [24]) (98.8% ± 1.3% versus 15.4% ± 2.9%, *p* < 0.01), and lower level of BAH-1 (a marker for human megakaryocyte-erythroid progenitor [22]) (0.51% ± 0.2% versus 32.5% ± 3.9%, *p* < 0.01) (Figure 1C,D). Due to co-expressions of CD7 and CD10 (the surface markers for common lymphoid progenitor (CLP) cells [25]) on miCD34^+^ HSCs, the data suggested that miCD34^+^ HSCs might have a high potential to give rise to lymphocytes.

### 2.2. Differentiation of miCD34^+^ HSCs into T Cells

To determine whether miCD34^+^ cells were functional as stem cells, they were purified from miPB-IPCs and treated with different inducers (Figure 2A). We first examined their potential to differentiate into T cells by treating purified miCD34^+^ cells with recombinant FMS-like tyrosine kinase (FLT)-3 ligand, interleukin (IL)-2, and IL-7 for 3 days. Phase-contrast microscopy revealed marked morphological changes, and the differentiated T cells had numbers of cell clusters in this cytokine-treated group, with some cells released into the supernatant (Figure 2B, right). Cells in the control groups exhibited a smooth surface and failed to show any morphological changes (Figure 2B, left, and Figure 2C, left). Confocal microscopy demonstrated that the differentiated cells strongly expressed human T cell marker CD4, with weak expression of CD8 (Figure 2D). Flow cytometry further confirmed the differentiation of miCD34^+^ HSCs into CD3^+^CD4^+^CD8^−^CD38^+^ T cells at a percentage of 76.93% ± 3.21% (Figure 2E, *n* = 4), which were CD3^+^CD4^+^TCRαβ^+^ T cells (82.65% ± 5.2%, *n* = 3) (Figure 2F). Intracellular staining with T-cell functional markers indicated that these T cells produced Th1 cytokine IL-12 (65.3% ± 20.1%, *n* = 3) and Th2 cytokines IL-4 (28.5% ± 9.99%, *n* = 3) and IL-5 (53.9% ± 11.2%, *n* = 3), with a very low level of interferon (IFN)-γ (0.61% ± 0.3%, *n* = 3) (Figure 2G). Additional functional tests established the significantly upregulated expression levels of cytokines such as IL-4 (*p* = 0.0025, *n* = 3), IL-5 (*p* = 0.0049, *n* = 3), and IL-12 (*p* = 0.037, *n* = 3) after the treatment with phorbol 12-myristate 13-acetate (PMA) and ionomycin. The level of INF-γ failed to show a marked change (*p* = 0.085, *n* = 3). The data confirmed that the differentiated T cells were responding to the stimulation of PMA/ionomycin (Figure 2H). The rapid differentiation of T cells with high efficiency clearly demonstrated that miCD34^+^ HSCs were converted into functional and definitive hematopoietic progenitors.

### 2.3. Ex Vivo Differentiation of miCD34^+^ HSCs into Other Hematopoietic Lineages

To further examine their potential for differentiation, we treated miCD34^+^ HSCs with macrophage colony-stimulating factor (M-CSF) for 3 days, whereupon they became adherent, well-spaced cells. Functional analysis established that M-CSF-treated miCD34^+^ HSCs exhibited strong phagocytosis of fluorescent latex beads (Figure 3A, middle), while untreated cells were mostly negative for this effect (Figure 3A, left). Flow cytometry confirmed that 34.3% ± 4.3% of the cells were CD11b^+^CD209^+^ macrophages (MΦ), and about 53.66% ± 3.8% were CD11b^+^CD209^−^ macrophages (MΦ) (Figure 3A, right). In contrast, there were only 15.29% ± 1.5% CD11b^+^CD209^−^ macrophages and 0.43% ± 0.12% CD11b^+^CD209^+^ macrophages in untreated miCD34^+^ HSCs.

Next, we induced miCD34^+^ HSCs to granulocyte colony-stimulating factor (G-CSF). After 3 days, 81.14% ± 3.7% of treated cells displayed the granulocyte-specific marker CD66b, with a reduced nuclear–cytoplasmic ratio and multi-lobed nuclei shown by Wright–Giemsa staining (Figure 3B). However, the untreated miCD34^+^ HSCs failed to express CD66b and displayed large nuclei, with a large nuclear–cytoplasmic ratio (Figure 3B, *n* = 4). Moreover, after being treated with erythropoietin (EPO) for 5 days, miCD34^+^ HSCs turned into nucleated cells strongly positive for the erythroid (Er) lineage marker CD235a and facilitated RBC maturation via expulsion of their nuclei with additional EPO treatment, exhibiting a distinctive biconcave shape and enucleated RBCs (Figure 3C, right). In total, 41.4% ± 11.46% of cells were terminally differentiated into enucleated CD235a^+^CD45^−^hemoglobin^+^ RBCs (Figure 3D, *n* = 4). However, untreated cells failed to show these differentiations, or only expressed background levels of these markers. Further flow cytometry analysis demonstrated that the level of hemoglobin expression was increased in the matured RBCs, with the mean fluorescence intensity of hemoglobin^+^CD45^−^ mature RBCs at 13.61 ± 4.29, while hemoglobin^+^CD45^+^ immature RBCs was 8.29 ± 1.61 (*p* = 0.044, *n* = 4).

Additionally, we examined the commitment of miCD34^+^ HSCs to MKs and platelets, which are critical for blood clotting. After treatment with FLT-3 ligand and thrombopoietin (TPO) for 7 days, production of CD42^+^ MKs was achieved with typical polyploidization (mostly from 2N to 7N) (Figure 3E–G) and the formation of non-nucleated CD42^+^ platelets (21.3% ± 4.1%, *n* = 4) (Figure 3E,F), yielding 95 ± 17 platelets per MK. Approximately 54% of mature CD42^+^ platelets were released into the supernatant (Figure 3F, *n* = 4).

### 2.4. In Vivo Differentiation of miCD34^+^ HSCs into Other Hematopoietic Lineages after Transplant into NSG Mice

To further demonstrate their multipotent features, the purified miCD34^+^ HSCs were transplanted into irradiated nonobese diabetic (NOD)/Lt-*scid*/*IL2Rγ*^null^ (NSG) mice (Appendix A). We examined the chimerism of human CD45^+^ cells in peripheral blood, spleen, and bone marrow of miCD34 HSC-engrafted mice at 12 weeks, by using flow cytometry analysis with blood cell lineage-specific markers [11,26] including T cells (CD3^+^CD4^+^), B cells (CD19^+^), monocytes (CD14^+^), granulocytes (CD66b^+^), erythroid cells (CD235a^+^), and megakaryocytes/platelets (CD41b^+^). We found that the engraftment levels of human CD45^+^ cells in blood (9.93% ± 9.62%, *p* = 0.035) and spleen (25.37% ± 21.89%, *p* = 0.018) were much higher than that in bone marrow (0.33% ± 0.15%) at 12 weeks post-transplantation (Figure 4A, *n* = 6 mice). The miCD34^+^ HSC-derived CD3^+^CD4^+^ T cells remained a predominant population at 57.92% ± 8.49% of human CD45^+^ blood cells at 12 weeks after transplantation, with different proportions of other engrafted cells in the blood. Similar data (59% ± 13.55% of CD3^+^CD4^+^ T cells) were obtained from splenocytes of miCD34^+^ HSC-engrafted mice (Figure 4B,C, *n* = 5 mice). By comparison, the percentage of miCD34^+^ HSC-derived CD3^+^CD4^+^ T cells in bone marrow was 49.45% ± 14.01% at 12 weeks, of CD19^+^ B cells was 5.24% ± 2.68%, and of CD41b^+^ megakaryocytes/platelets was 4.3% ± 2.0%, with CD14^+^ monocytes at 1.71% ± 2.36%, CD66b^+^ granulocytes at 0.51% ± 0.46%, CD235a^+^SYTO60^+^ nucleated erythroid cells at 0.22% ± 0.13%, and CD235a^+^SYTO60^−^ enucleated erythroid cells at 0.08% ± 0.05% (Figure 4D, *n* = 6 mice). The percentage of CD14^+^ monocytes was 2.1% ± 1.81% in the peripheral blood of miCD34^+^ HSC-transplanted mice. Additional flow cytometry failed to detect the primary phenotype of undifferentiated miCD34^+^ HSCs. There were few to no human CD34^+^ cells in peripheral blood (0.07% ± 0.04%), spleen (0.04% ± 0.03%), or bone marrow (0.01% ± 0.01%) of miCD34 HSC-engrafted mice at 12 weeks (Figure 4E, *n* = 5 mice).

To further confirm miCD34^+^ HSCs giving rise to monocytes/macrophages (myeloid lineage differentiation), we performed an additional animal study in miCD34^+^HSC-transplanted mice at 12 and 16 weeks, respectively, by using macrophage-associated markers anti-human CD11b and CD11c mAbs. Flow cytometry demonstrated that the percentage of mCD45^−^hCD45^+^hCD3^−^hCD11b^+^hCD11c^+^ macrophages was 75.22% ± 18.33% in the splenocytes of miCD34^+^ HSC-transplanted mice at 16 weeks (Figure 4F, *n* = 3). In contrast, the percentage of mCD45^−^hCD45^+^hCD3^+^hCD11b^−^ T cells declined from 59% ± 14.01% at 12 weeks to 4.05% ± 2.87% at 16 weeks (Figure 4F). Thus, the data clearly demonstrated the monocyte/macrophage (myeloid) differentiation of miD34^+^ HSCs after transplantation into the irradiated NSG mice. Considering other lineage differentiations (e.g., T cells, B cells, megakaryocytes and red blood cells), these data indicated the multi-lineage differentiations of miCD34^+^ HSCs.

### 2.5. Notch Signaling Pathway Contributed to the miCD34^+^ HSC Differentiation after Treatment with Platelet-Derived Mitochondria

Notch signaling has been well established as an essential regulator for HSC generation and differentiation [27]. Specifically, the Notch signaling pathway plays a crucial role in T-cell development and maturation at different stages [28]. Both ex vivo and in vivo data demonstrated the multiple differentiations of miCD34^+^ HSCs. To dissect the molecular mechanisms underlying mitochondrial treatment, we explored the action of Notch signaling during the induction of differentiation of PB-IPCs toward the miCD34^+^ HSCs. Flow cytometry revealed that mitochondria expressed Notch ligands Jagged 1 (JAG1) (25.13% ± 16.0%), Jagged 2 (JAG2) (68.04% ± 14.6%) and Delta-like 3 (DLL3) (69.3% ± 25.96%); DLL1 (2.21% ± 1.74%) and DLL4 (0.23% ± 0.09%) (Figure 5A) were not expressed. The expression levels of Notch receptors 1 – 4 on PB-IPCs were markedly upregulated after the treatment with platelet-derived mitochondria (Figure 5B). To examine the role of Notch signaling in miCD34^+^ HSC differentiation of mitochondrion-induced PB-IPCs, N-[N-(3,5-difluorophenacetyl)-L-alanyl]-S-phenylglycine t-butyl ester (DAPT) treatment was used to block γ-secretase (Figure 5C), an enzyme critical for the release of the Notch intracellular domain (NICD) into the nucleus to initiate gene transcription [29]. The percentage of CD34^+^ cells was significantly increased in the group treated with mitochondria + DAPT (26.2% ± 5.68%) (Figure 5D). Contrastingly, treatment with DAPT alone showed a very low ability to induce CD34^+^ cells (2.59% ± 0.13%), indicating that mitochondria are required for miCD34^+^ HSC cell differentiation.

## 3. Discussion

The use of autologous stem cells for regenerative medicine is more ethically acceptable and likely to be more successful than the use of other stem cells, and such therapies would avoid many of the immune rejection and safety concerns associated with other stem cells (e.g., ES- or iPS-based therapies). PB-IPCs can be easily isolated from peripheral blood and expanded in serum-free culture medium to avoid the painful and invasive procedures required to withdraw bone marrow. Using autologous PB-IPCs from patients as a starting material, mitochondrial treatment can generate functional autologous miCD34^+^ HSCs on a large scale, giving rise to different blood cell lineages such as T cells, MΦ, granulocytes, erythrocytes, and MKs/platelets. Current data indicated that miCD34^+^ HSCs exhibited a rapid multiple potential for differentiations post-engraftment into irradiated NSG mice. Thus, these cells offer great promise as a solution for the current bottlenecks associated with conventional stem-cell transplants and have tremendous potential for patient benefit in the clinic. Generation of functional autologous miCD34^+^ HSCs from PB-IPCs may address an unmet medical need.

Both ex vivo and in vivo studies demonstrated the T-cell differentiation of miCD34^+^ HSCs at high efficiency. To exclude the possibility of the expansion of contaminated cells, we carefully designed experiments and analyzed the differentiated cells through morphology and flow cytometry with multiple different T-cell specific markers such as CD3, CD4, CD8, TCRα/β, TCRγ/δ, and Th1 and Th2 cytokines. There were a few CD8^+^ T cells adhered to PB-IPCs after overnight (12 h) culture in non-tissue-culture-treated Petri dishes, but no CD4^+^ T cells. After the expansion of PB-IPCs for 7 days, PB-IPCs were attached to the bottom of the Petri dish with high purity (>97%) and remained negative for CD4, with a few CD8^+^ T cells (~2.63%). Before the treatment with platelet-derived mitochondria, all floating cells and cellular debris were washed away twice with PBS. The purified miCD34^+^ HSCs were utilized for transplantation. Thus, the possibility of transplanting mature PBMCs and/or lymphocytes was very limited in our protocol. Normally, NSG mice fail to support human T-cell growth and differentiation. A few CD8^+^ T cells might be temporarily expanded after transplant in mice; however, flow cytometry demonstrated that most of the T cells were CD3^+^CD4^+^ T cells in the miCD34^+^-HSC-transplanted NSG mice. Therefore, the engrafted CD3^+^CD4^+^ T cells were differentiated from miCD34^+^ HSC. Additionally, our ex vivo studies demonstrated the high efficiency of CD3^+^CD4^+^CD8^−^CD38^+^ T-cell differentiation from miCD34^+^ HSCs, with an expression of Th1- and Th2-associated cytokines (IL-4, IL-5, and IL-12). To confirm that the CD4 expression was on T cells, not on monocytes/macrophages, the T-cell-specific marker CD3 was utilized for cellular gating during the flow cytometry. Therefore, the CD4^+^ T cells were derived from the differentiation of miCD34^+^ HSCs.

Notch receptors act as key regulators in both CD34^+^ HSC and T-cell development and maturation at different stages [27,28]. After binding to its specific ligands, the Notch intracellular domain (NICD) is cleaved off by the enzyme γ-secretase and translocated to the nucleus, where it binds to the transcription factor recombination signal sequence-binding protein Jkappa (RBP-J), leading to gene regulation. To understand the mechanism underlying miCD34^+^ HSC and T-cell differentiations, the current study revealed the upregulation of Notch receptors 1–4 on miPB-IPC, while platelet-derived mitochondria expressed Notch ligands Jagged 1 (JAG1), JAG3 and Delta-like 3 (DLL3), but not DLL1 and DLL4, highlighting that the Notch signaling pathway may contribute to the miCD34^+^ HSC and T-cell differentiations after treatment with platelet-derived mitochondria. Notably, the percentage of miCD34^+^ HSCs was significantly upregulated after mitochondrial treatment with addition of the γ-secretase inhibitor DAPT. The data confirmed that the canonical Notch pathway contributed to the differentiation of miCD34^+^ HSCs through the interaction of Notch receptors on PB-IPCs and their ligands on platelet-derived mitochondria. Due to the action of the Notch pathway in the maintenance of quiescent feature of hematopoietic stem and progenitor cells [30], blocking with DAPT promoted the miCD34^+^ HSC differentiation of PB-IPCs, which was consistent with a previous report [30]. Additional detailed molecular mechanisms need to be explored to better understand the individual and synergistic effects during the interaction of mitochondrial Notch ligands (JAG1, JAG3, and DLL3) with Notch receptors 1–4 on PB-IPCs.

The roadmap of cell differentiation and maturation is tightly modulated through the systematic integration of distinct activating or repressing signaling pathways located both in the cytoplasm and nucleus, specific to the hematopoietic system, underlying the hierarchical differentiation model [22,26,31]. Our previous work [18] demonstrated that highly purified adult peripheral blood (PB)-derived platelets (>99% purity) strongly displayed ES cell-associated pluripotent gene markers such as transcription factors OCT3/4 and SOX2, with little or no expression of NANOG. Real-time PCR array revealed the expressions of human-stem-cell-related transcription factors and human-stem-cell-associated markers in the mitochondria of human PB-platelets [18], highlighting that the stemness markers are localized in platelets’ mitochondria, which may contribute to the induction of multipotency of PB-IPCs after treatment with PB-derived mitochondria. Our recent mechanistic studies confirmed that mitochondria enter cells and directly penetrate the nuclei of PB-IPCs after treatment with platelet-derived mitochondria, where they can produce profound epigenetic changes [21]. Due to the essential role of mitochondria in the reprogramming of somatic cells to iPS cells [32], additional molecular mechanisms underlying the mitochondrial reprogramming of PB-IPCs need to be determined. In conclusion, innovative reprogramming of adult PB-IPCs by treatment with platelet-derived mitochondria may overcome the limitations and safety concerns associated with using conventional transgenic technologies to reprogram ES and iPS cells in the clinical setting.

## 4. Materials and Methods

### 4.1. PB-IPC Cell Culture

Human buffy coat blood units (*N* = 51; mean age of 48.97 ± 14.11; age range from 18 to 72 years old; 24 males and 27 females) were purchased from the New York Blood Center (New York, NY, USA, http://nybloodcenter.org/). Human buffy coats were initially added to 40 mL chemical-defined serum-free culture X-VIVO 15^TM^ medium (Lonza, Walkersville, MD, USA) and mixed thoroughly with a 10 mL pipette, and then used for isolation of peripheral-blood-derived mononuclear cells (PBMCs). PBMCs were harvested as previously described [33]. Briefly, mononuclear cells were isolated from buffy coat blood using Ficoll–Paque^TM^ PLUS (γ = 1.007, GE Healthcare, Chicago, IL, USA), followed by removing the red blood cells using red blood cell lysis buffer (eBioscience, San Diego, CA, USA). After three washes with saline, the whole PBMCs were seeded in 150 × 15 mm Petri dishes (BD Falcon, NC, USA) at 1 × 10^6^ cells/mL, 25 mL/dish in chemical-defined serum-free culture X-VIVO 15^TM^ medium (Lonza, Walkersville, MD, USA) without any other added growth factors, and incubated at 37 °C in 8% CO_2_ [34]. Seven days later, PB-IPCs were growing and expanded by adhering to the hydrophobic bottom of Petri dishes. Subsequently, PB-IPCs were washed three times with saline and all floating cells were removed. Next, the serum-free NutriStem^®^ hPSC XF culture medium (Corning) was added to continue cell culture and expansion at 37 °C in 8% CO_2_. The expanded PB-IPCs were usually applied for experiments within 7–14 days.

### 4.2. Isolation of Mitochondria from Platelets

The mitochondria were isolated from PB-platelets using the Mitochondria Isolation kit (Thermo Scientific, Rockford, IL, USA, Prod: 89874) according to the manufacturer’s recommended protocol [18]. Adult human platelet units (*N* = 16; mean age of 30.81 ± 8.64; age range from 16 to 40 years old; 9 males and 7 females) were purchased from the New York Blood Center (New York, NY, USA, http://nybloodcenter.org/). The concentration of mitochondria was determined by the measurement of protein concentration using a NanoDrop 2000 Spectrophotometer (ThermoFisher Scientific, Waltham, MA, USA). The isolated mitochondria were aliquoted and kept in a −80 °C freezer for experiments.

For mitochondrial staining with fluorescent dyes, mitochondria were labeled with MitoTracker Deep Red FM (100 nM) (Thermo Fisher Scientific, Waltham, MA, USA) at 37 °C for 15 min according to the manufacturer’s recommended protocol, followed by two washes with PBS at 3000 rpm × 15 min [18].

### 4.3. In Vitro Differentiation of PB-IPCs into miCD34^+^ HSCs

PB-IPCs were treated with 100 μg/mL platelet-derived mitochondria for 7–14 days in the non-treated 24 well plates or Petri dishes with the serum-free NutriStem^®^ hPSC XF culture medium (Corning, New York, NY, USA), at 37 °C and 8% CO_2_. According to our current protocol, the miCD34^+^ HSCs were purified from mitochondria-treated PB-IPCs by immunomagnetic sorting with Miltenyi Biotech CD34 MicroBead Kit (Miltenyi Biotech, Gladbach, Germany, catalog #130-097-047) according to the manufacturer’s instructions. The differentiation of miCD34^+^ HSCs was characterized by flow cytometry.

### 4.4. Flow Cytometry

Flow cytometric analyses of surface and intracellular markers were performed as previously described [18]. PB-IPCs were washed with PBS at 2000 rpm for 5 min. Mitochondria were washed with PBS at 12,000× *g* for 10 min at 4 °C. Samples were pre-incubated with human BD Fc Block (BD Pharmingen, Franklin Lakes, NJ, USA) for 15 min at room temperature, and then directly aliquoted for different antibody staining. Cells were incubated with different mouse anti-human monoclonal antibodies (mAb). For surface staining, cells were stained for 30 min at room temperature and then washed with PBS at 2000 rpm for 5 min prior to flow analysis. Isotype-matched mouse anti-human IgG antibodies (Beckman Coulter, Brea, CA, USA) served as a negative control for all fluorescein-conjugated IgG mAb. SYTOTM60 (Thermo Fisher, Waltham, MA, USA) was combined with CD235a (GLY-A) staining to determine the nucleated erythroid cells. Staining with propidium iodide (PI) (BD Biosciences, San Jose, CA, USA) was used to exclude dead cells during the flow cytometry analysis. For intracellular staining, cells were fixed and permeabilized according to the PerFix-nc kit (Beckman Coulter) manufacturer’s recommended protocol. After staining, cells were collected and analyzed using a Gallios Flow Cytometer (Beckman Coulter, Brea, CA, USA) equipped with three lasers (488 nm blue, 638 red, and 405 violet lasers) for the concurrent reading of up to 10 colors. The final data were analyzed using the Kaluza Flow Cytometry Analysis software version 2.1 (Beckman Coulter).

Cells were incubated with different mouse anti-human monoclonal antibodies (mAb) from Beckman Coulter (Brea, CA, USA) including FITC-conjugated anti-CD45RA, anti-IFNγ, anti-CD4, anti-CD235a, anti-CD8 and anti-CD42a; phycoerythrin (PE)-conjugated anti-CD34; PE-Texas Red-conjugated CD3; phycoerythrin-Cy5 (PE-Cy5)-conjugated anti-CD90; phycoerythrin-Cy5.5 (PE-Cy5.5)-conjugated anti-CD19; phycoerythrin-Cy7 (PE-Cy7)-conjugated anti-CD49f, anti-CD11b and anti-CD45; APC-conjugated anti-CD4; APC-Alexa Fluor 700-conjugated anti-CD71; APC-Alexa Fluor 750-conjugated anti-CD7, anti-CD66b and anti-CD8; Pacific blue (PB)-conjugated anti-CD38; Krome Orange-conjugated anti-CD14 and anti-135 (FLT3) -BV510. From BD Biosciences (San Jose, CA, USA), the investigators purchased the AlexaFluor-488-conjugated anti-human Cytochrome C; FITC-conjugated anti-CD90 (THY1) and anti-CD11C; BV 510-conjugated anti-CD45, PE-conjugated anti-IL4, anti-IL5, anti-BAH1, anti-IL12; PE-CF594-conjugated anti-CD10, BV421-conjugated anti-CD209 and PE-conjugated anti-mouse CD45.1. Antibodies were purchased from Biolegend (San Diego, CA, USA) including the FITC-conjugated anti-human Hsp60, anti-human TCRαβ, anti-human Notch 1 and anti-human Notch 2; PE-conjugated anti-human Notch 3, anti-human Dll1, anti-human Dll4 and anti-human Jagged2;APC conjugated anti-human Notch 4; phycoerythrin-Cy7 (PE-Cy7)-conjugated anti-TCRγδ and Pacific blue (PB)-conjugated anti-CD3. FITC-conjugated anti-human Jagged1 and PE-conjugated anti-human Dll3 mAbs were purchased from R&D Systems (Minneapolis, MN, USA). Hemoglobinβ/γ/δ (H-76) rabbit polyclonal antibody, AlexaFluor-546-conjugated Calnexin and AlexaFluor-647-conjugated GM130 were purchased from Santa Cruz Biotechnology (Dallas, TX, USA). The SYTOTM60 was purchased from Thermo Fisher (Waltham, MA, USA).

### 4.5. Multiple Differentiations of miCD34^+^ HSCs

Initially, miCD34^+^ HSCs were purified from miPB-IPCs by using CD34 MicroBead Kit, human-lyophilized (Miltenyi Biotec, Gladbach, Germany) through the auto MACS Pro Separator (Miltenyi Biotec, Gladbach, Germany) according to manufacturer’s recommended protocol. The purified miCD34^+^ HSCs were treated with different inducers for cellular differentiations.

To test T-cell differentiation, the purified miCD34^+^ HSCs (1 × 10^5^ cells/mL) were planted in 24 well non-treated plates in the presence of HSC-Brew GMP Basal Medium (Miltenyi Biotec, Gladbach, Germany) with addition of cytokines 25 ng/mL hFlt3L and 25 ng/mL rhIL -7 (R&D Systems, Minneapolis, MN, USA), at 37 °C in 5% CO_2_. After the treatment for 3–7 days, cells were photographed and analyzed by confocal microscopy and flow cytometry using different T-cell markers such as CD3, CD4, CD8, TCR α/β, CD38, Th1 cytokines (IL-4 and IL-5) and Th2 cytokines (IFN-γ and IL-12). Untreated miCD34^+^ HSCs served as negative controls. T cells from healthy donors served as positive controls. For combined immunocytochemistry, the differentiated cells were fixed in 24 well plates with 4% paraformaldehyde for 20 min and permeabilized with 0.5% triton X-100 (Sigma, Saint Louis, MO, USA) for 5 min, blocking non-specific binding with 2.5% horse serum, and followed by immunostaining with FITC-conjugated mouse anti-human CD4 and CD8 (Beckman Coulter, Brea, CA, USA). After covering with mounting medium with DAPI (Vector Laboratories, Burlingame, CA, USA), cells were photographed with a Nikon A1R confocal microscope on a Nikon Eclipse Ti2 inverted base, using NIS Elements Version 4.60 software.

For the differentiation of miCD34^+^ HSCs to macrophages, the purified miCD34^+^ HSCs (1 × 10^5^ cells/mL) were treated with 50 ng/mL M-CSF (Sigma, St. Louis, MO, USA) in 24 well non-treated plates in the presence of HSC-Brew GMP Basal Medium, at 37 °C in 5% CO_2_. After treatment for 2–3 days, cells were analyzed with phagocytosis and by flow cytometry with macrophage marker CD11b (Beckman Coulter, Brea, CA, USA) and CD209 (BD Biosciences, San Jose, CA, USA). Untreated miCD34^+^ HSCs served as negative controls. To detect the function of differentiated macrophages, fluorescence latex beads (Sigma, Saint Louis, MO, USA) were added to M-CSF-treated and untreated miCD34^+^ HSC cultures. After 4 h of incubation with latex beads, cells were washed three times with PBS. The phagocytosis was viewed and evaluated under microscopy. The positive cells had a minimum of five beads per cell.

To differentiate the miCD34^+^ HSCs into granulocytes, the purified miCD34^+^ HSCs (1 × 10^5^ cells/mL) were treated with 25 ng/mL hFlt3L + 100 ng/mL G-CSF (R&D Systems) in the presence of HSC-Brew GMP Basal Medium, in the 24 well non-treated plates, at 37 °C in 5% CO_2_. After the treatment for 3–5 days, cells were photographed and analyzed by flow cytometry with granulocyte marker CD66b and staining with Wright–Giemsa (Sigma, Saint Louis, MO, USA) according to the manufacturer’s instructions. Untreated miCD34^+^ HSCs served as negative controls. PBMCs from healthy donors served as positive controls.

To differentiate miCD34^+^ HSCs into RBCs, purified miCD34^+^ HSCs (1 × 10^5^ cells/mL) were initially treated with 25 ng/mL hFlt3L + 3 units/mL EPO (R&D Systems, Minneapolis, MN, USA) in the presence of HSC-Brew GMP Basal Medium, in the 24 well non-treated plates, at 37 °C in 5% CO_2_. After this treatment for 5 days, cells were re-treated with 3 units/mL EPO for an additional 3–7 days. Subsequently, cells were photographed and analyzed by flow cytometry with erythrocyte markers CD235a and hemoglobin. Untreated miCD34^+^ HSCs served as negative controls. For intracellular flow cytometry, all floating cells were collected and centrifuged at 2700× *g* 15 min. First, after blocking non-specific binding with Fc Blocker (BD Biosciences, San Jose, CA, USA), cells were fixed and permeabilized using the PerFix-nc kit (Beckman Coulter, Brea, CA, USA) according to the manufacturer’s recommended protocol. Second, cells were incubated with rabbit anti-human hemoglobinβ/γ/δ polyclonal antibody (Santa Cruze, Dallas, TX, USA) at 1:100 dilution, room temperature for 30 min, and then washed with PBS at 2700× *g* 15 min. Next, cells were labeled with Cy5-conjugated AffiniPure donkey anti-rabbit 2nd Ab (Jackson ImmunoResearch Laboratories, West Grove, PA, USA), in combination with staining with mouse anti-human CD235a-FITC (Beckman Coulter, Brea, CA, USA) and CD45-PE-CY7 mAbs for 30 min, and followed by flow cytometry analysis.

To differentiate miCD34^+^ HSCs into megakaryocytes and platelets, the purified miCD34^+^ HSCs (1 × 10^5^ cells/mL) were initially treated with 25 ng/mL hFlt3L + 100 ng/mL TPO (R&D Systems, Minneapolis, MN, USA) in the presence of HSC-Brew GMP Basal Medium, in the 24 well non-treated plates, at 37 °C in 5% CO_2_. After this treatment for 3–7 days, cells were photographed and collected for flow cytometry with MK/platelet marker CD42a (Beckman Coulter, Brea, CA, USA). Untreated miCD34^+^ HSCs served as negative controls. For the analysis of polyploidization, viable TPO-treated miCD34^+^ cells were first stained with CD42a mAb and Hoechst 33342 (Sigma, Saint Louis, MO, USA) and photographed under a confocal microscope. Secondly, using healthy donor-derived matured T cells (1N) and platelets (0N) as controls, the polyploidy of differentiated CD42a^+^ MKs was analyzed by flow cytometry after staining with propidium iodide (PI) (Abcam, Cambridge, MA, USA) according to the manufacturer’s recommended protocol.

To morphologically determine the differentiation of miCD34^+^ HSCs to granulocytes, RBCs, and megakaryocytes/platelets, Wright–Giemsa staining was performed on the treated and untreated cells, which were then observed and photographed under an inverted Nikon ECLIPSE Ti2 microscope.

For the DAPT-blocking experiment, PB-IPCs were treated with 100 μg/mL mitochondria + 10 μM DAPT (Sigma, Saint Louis, MO, USA, Catalog# D5942) for 7–14 days in the non-tissue-culture-treated 24 well plates or Petri dishes with the serum-free NutriStem^®^ hPSC XF culture medium (Corning, New York, NY, USA), at 37 °C and 8% CO_2_. Subsequently, both treated and untreated PB-IPC were collected to examine the expression of CD34 by flow cytometry.

### 4.6. Animal Study and Engraftment of miCD34+ HSCs into Irradiated NSG Mice

All animal experiments were performed according to approval of the institutional Animal Care and Use Committee of Hackensack Meridian Health (approved on 14/12/2018 approval code: 259.00). To demonstrate the multipotent features of miCD34^+^ HSCs, the purified miCD34^+^ HSCs were transplanted into irradiated NOD/Lt-*scid*/*IL2Rγ*^null^ (NSG) mice [35]. NSG mice were purchased from Jackson Laboratories (Bar Harbor, ME, USA), and were bred and maintained under pathogen-free conditions at the animal facility of the Center for Discovery and Innovation. To determine the repopulating potential of miCD34^+^ HSCs, NSG mice, aged 12–16 weeks, were irradiated with 200 cGy using an RS-2000 irradiator (Rad Source Technologies, Suwanee, GA, USA). The miCD34^+^ HSCs were transplanted into irradiated NSG mice at 3 × 10^5^ cells/mouse via the tail vein (in 200 μL saline, i.v., *n* = 10 mice) 24 h after irradiation, according to a protocol approved by the Animal Care and Use Committee (ACC) of the Hackensack Meridian Health. Only physiological saline injection (200 μL) served as a control (*n* = 6 mice). After transplantation, mice were monitored twice a week for 16 weeks. To examine the differentiation of miCD34^+^ HSCs, mice were sacrificed at the 12 or 16 weeks post-transplantation to collect samples of peripheral blood, spleen, and bone marrow for flow cytometry analysis. To determine the multilineage differentiation of miCD34^+^ HSCs after transplantation into irradiated NSG mice, only the viable cells from different samples were gated for analysis after excluding the propidium iodide (PI)-positive dead cells. The gated human leukocyte common antigen CD45-positive and mouse CD45.1-negative viable cells were further analyzed for characterization with different human blood-cell-lineage-specific surface markers such as CD3 and CD4 for T cells; CD19 for B cells; CD41b for megakaryocytes/platelets; CD14, CD11b, and CD11c for monocytes/macrophages; CD66b for granulocytes; and CD235a for erythroid cells. SYTO60 was utilized to stain the CD235a^+^ nucleated erythroid cells. To determine T-cell populations and remove CD4^+^ monocytes, anti-CD3 Ab was employed to gate out CD4^+^ monocytes, in addition to the consideration of cell-size difference. Isotype-matched IgGs served as controls for flow cytometry.

### 4.7. Statistics

Statistical analyses were performed with GraphPad Prism 8 (version 8.0.1) software. The normality test of samples was evaluated using the Shapiro–Wilk test. Statistical analyses of data were performed using the two-tailed paired Student’s *t*-test to determine statistical significance between untreated and treated groups. The Mann–Whitney U test was utilized for non-parametric data. Values are given as mean ± SD (standard deviation). Statistical significance was defined as *p* < 0.05, with two sided.

## Figures and Tables

**Figure 1 ijms-21-04249-f001:**
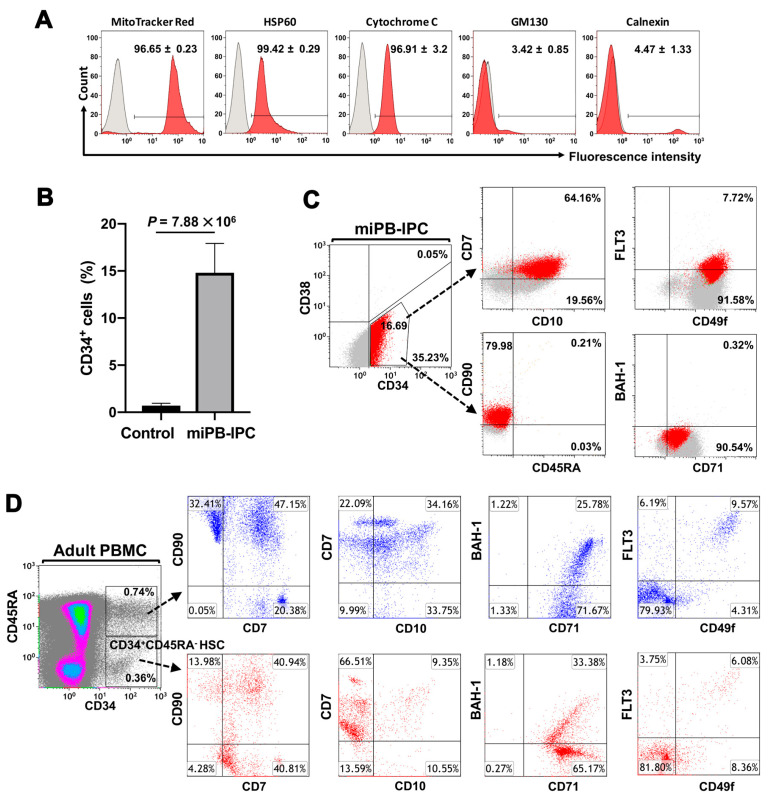
Differentiation of PB-IPCs into CD34^+^ HSC-like cells after their treatment with platelet-derived mitochondria. (**A**) The purity analysis of isolated mitochondria. The different markers were applied by flow cytometry, including MitoTrack Deep Red staining, anti-cytochrome C, and anti-heat shock protein (HSP) 60 Abs for mitochondrial markers, calnexin for endoplasmic reticulum (ER), and GM130 for Golgi apparatus. Isotype-matched IgGs (grey histogram) served as negative controls (*n* = 3). (**B**) CD34 expression upregulation after treatment with mitochondria in miPB-IPCs. Data represent mean ± SD of five experiments. (**C**) Phenotypic characterization of gated miCD34^+^ HSCs (dotted arrows) with additional surface markers (red) in total miPB-IPCs. Isotype-matched IgGs served as controls. Data were representative from five preparations. (**D**) Phenotypic characterization of gated CD34^+^CD45RA^−^ HSCs (dotted arrow) with additional markers (bottom, red) and CD34^+^CD45RA^+^ cell population (dotted arrow) with additional markers (top, blue) in total PBMCs (*n* = 4). Isotype-matched IgGs served as controls. Data were representative from one of four preparations.

**Figure 2 ijms-21-04249-f002:**
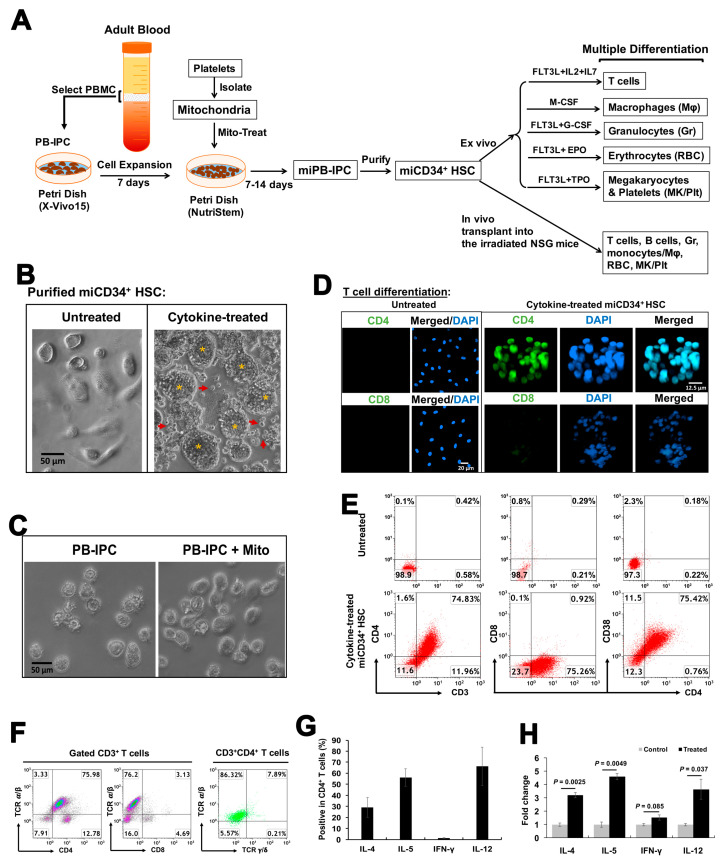
In vitro differentiation of purified miCD34^+^ HSCs to T cells. (**A**) Outline of the whole protocol from the generation of miPB-IPCs to multipotent cellular differentiations of miPB-IPCs. The miCD34^+^ HSCs were purified for in vitro and in vivo differentiations, respectively. (**B**) Phase-contrast images showing T-cell differentiation of purified miCD34^+^ HSCs in the presence of FLT-3 ligand, IL-2, and IL-7 for 3 days (*n* = 4). Untreated cells served as a control (left). The treated-CD34^+^ HSCs displayed substantial morphological changes with cell clusters (orange stars), and some floating cells (red arrows) released from “cell clusters” (orange stars). Magnification ×200. (**C**) Phase-contrast images show the morphology of control PB-IPCs (left) and treated PB-IPCs in the presence of mitochondria (right) (*n* = 4). Original magnification: ×200. (**D**) Z-stacked confocal images demonstrated strong expression of human T-cell marker CD4 with a low expression of CD8 (*n* = 4). Untreated miCD34^+^ HSCs served as control for immunostaining (left panel). (**E**) Flow cytometry revealed that differentiated T cells were CD3^+^CD4^+^CD8^−^CD38^+^ (*n* = 4). Untreated miCD34^+^ HSCs served as controls (top panel). (**F**) Expression of T-cell receptors α/β (TCRαβ) in the gated CD3^+^ and CD4^+^ T cells (*n* = 4). (**G**) Intracellular staining of differentiated T cells with Th1/Th2 cell cytokine markers (*n* = 3). Isotype-matched IgG served as a control. Data are presented as mean ± SD from three experiments. (**H**) Fold-changes of cytokine expression levels in the in vitro differentiated T cells after the stimulation with PMA and ionomycin (*n* = 3). Data represent mean ± SD.

**Figure 3 ijms-21-04249-f003:**
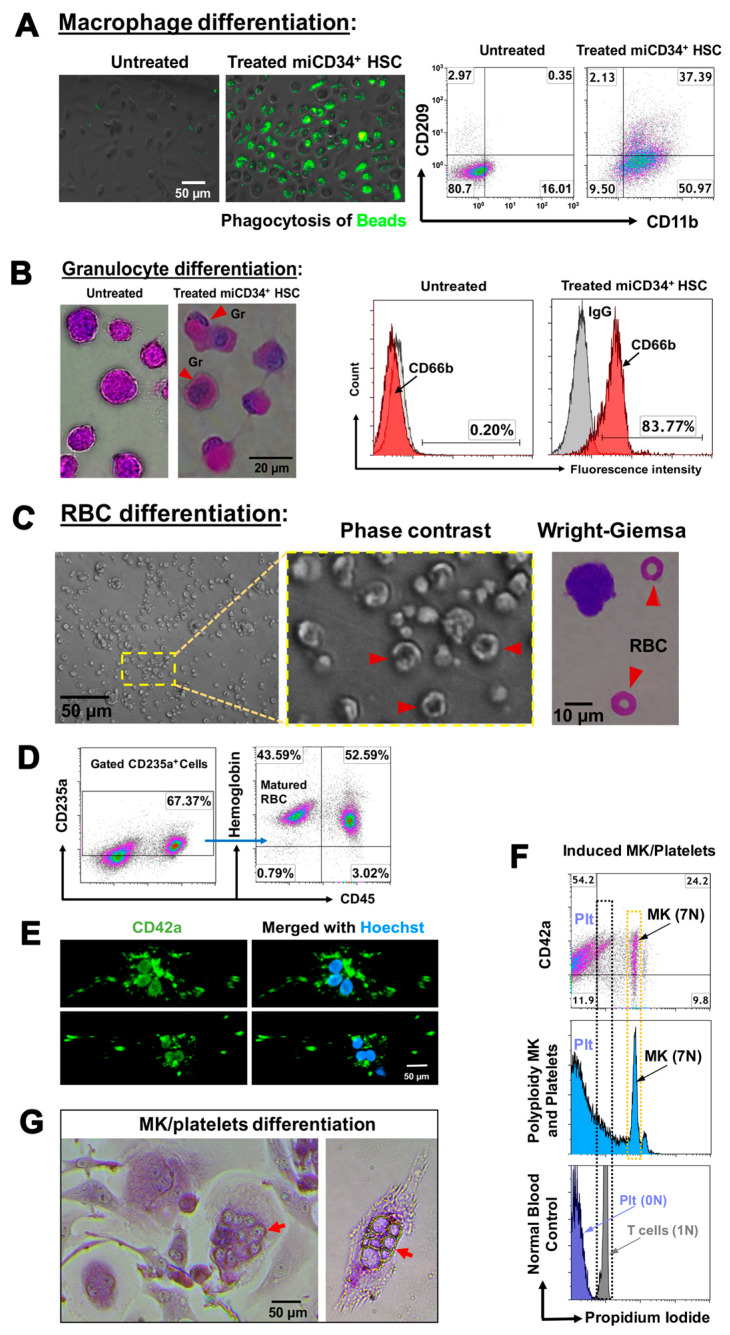
Ex vivo multiple differentiations of miCD34^+^ HSCs. (**A**) Differentiation of miCD34^+^ HSCs into macrophages after treatment with 50 ng/mL M-CSF for 3 days. The differentiated Mφ displayed macrophage markers such as CD11b and CD209, and exhibited phagocytosis of florescence beads (green) (*n* = 4). Untreated miCD34^+^ HSCs served as controls. (**B**) Differentiation of miCD34^+^ HSCs into granulocytes after treatment with 100 ng/mL G-CSF + 25 ng/mL FLT-3L for 3 days, followed by Wright–Giemsa staining (left), and flow cytometry for granulocyte marker CD66b (*n* = 4). Untreated miCD34^+^ HSCs served as controls. (**C**) Differentiation of miCD34^+^ HSCs into erythrocytes shown by phase-contrast imaging of mature RBCs (indicated by red arrow), and by Wright–Giemsa staining with typical morphology of mature RBCs (indicated by red arrows) (*n* = 4). (**D**) Analysis of the percentage of the matured CD45^−^hemoglobin^+^ RBCs in the gated CD235a^+^ cells by flow cytometry (*n* = 4). (**E**) Differentiation of miCD34^+^ HSCs into megakaryocytes (MKs)/platelets after the treatment with FLT-3L + TPO for 7 days, exhibiting CD42^+^ and a polynuclear appearance (*n* = 4). (**F**) Analysis of polyploid MKs post-treatment with FLT-3L + TPO for 7 days, shown by histogram (blue, middle) and dot plot with MK/platelet marker CD42a (top). Normal platelets (dark blue) and T cells (grey) from healthy donors served as controls for cells with no nucleus (0N) and one nucleus (1N), respectively (bottom) (*n* = 4). (**G**) Wright–Giemsa staining showed the differentiated miCD34^+^ HSCs with multiple nuclei (indicated by red arrows) after the treatment with FLT-3L + TPO for 7 days (*n* = 4).

**Figure 4 ijms-21-04249-f004:**
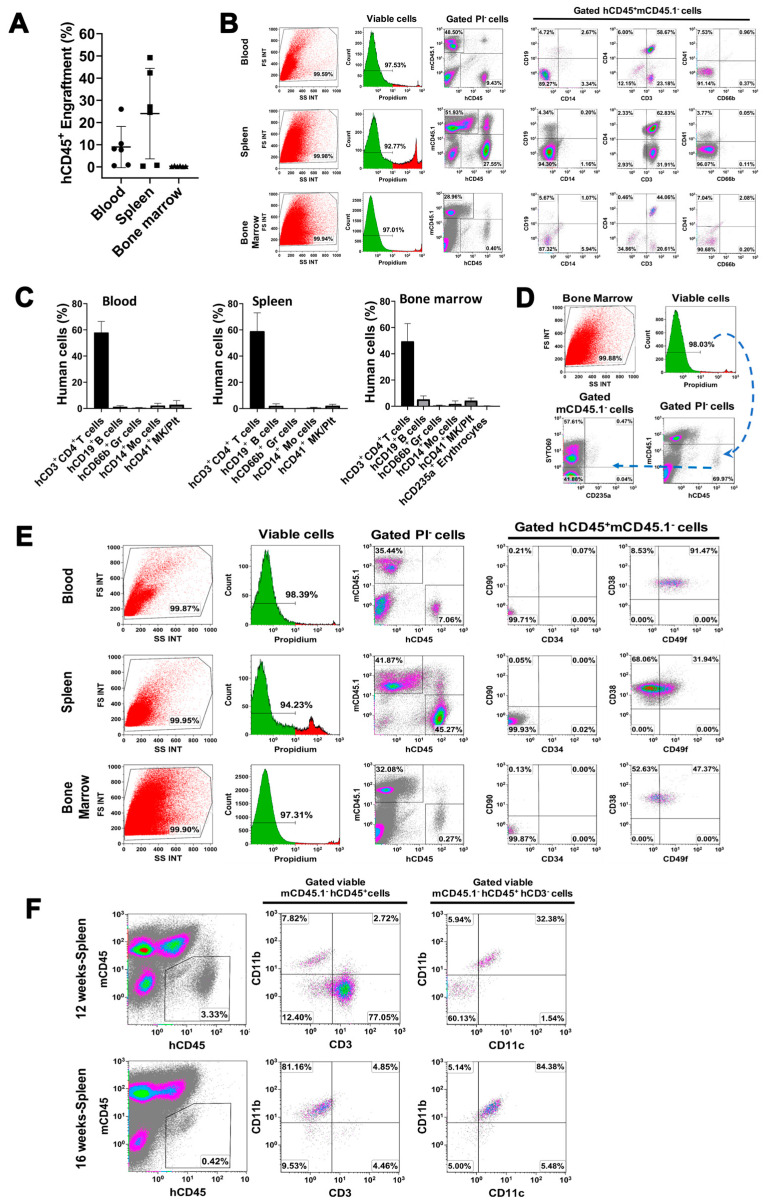
In vivo multiple differentiations of miCD34^+^ HSCs after transplantation into the irradiated NSG mice. To determine the multilineage differentiations of miCD34^+^ HSCs after transplantation into irradiated NSG mice, only the viable cells from different samples were gated for analysis after excluding the propidium iodide (PI)-positive dead cells. The gated human leukocyte common antigen CD45-positive and mouse CD45.1-negative viable cells were analyzed for characterization with different lineage-specific surface markers such as CD3 and CD4 for T cells; CD19 for B cells; CD41b for megakaryocytes/platelets; CD14, CD11b, and CD11c for monocytes/macrophages; CD66b for granulocytes; and CD235a for erythroid cells. SYTO60 was utilized to stain the CD235a^+^ nucleated erythroid cells. Isotype-matched IgGs served as controls for flow cytometry. (**A**) Engrafted levels of hCD45^+^mCD45.1^−^ cells in peripheral blood, spleen, and bone marrow of miCD34^+^ HSC-transplanted NSG mice at 12 weeks (3 × 10^5^ cells/mouse in 200 μL physiological saline, i.v., *n* = 6). (**B**) Phenotypic characterization of miCD34^+^ HSCs post engraftment into the irradiated NSG mice (3 × 10^5^ cells/mouse in 200 μL physiological saline, i.v., *n* = 5). Tissue samples were collected 12 weeks after the transplantation. Only the viable cells (green histogram) were gated for analysis after excluding the propidium iodide (PI)-positive dead cells. The gated human leukocyte common antigen CD45-positive and mouse CD45.1-negative viable cells were analyzed. Representative data are from one of three experiments with similar results. Isotype-matched IgGs served as controls for flow cytometry. (**C**) Multi-lineage differentiations of miCD34^+^ HSCs at 12 weeks after transplantation into irradiated NSG mice. (**D**) Erythroid reconstitution of miCD34^+^-transplanted NSG mice at 12 weeks (*n* = 6). Only the viable cells (green histogram) were gated for analysis after excluding the propidium iodide (PI)-positive dead cells. SYTO60 was utilized to stain the CD235a^+^ nucleated erythroid cells [11]. (**E**) Characterization of miCD34^+^ HSCs 12 weeks after engraftment in the irradiated NSG mice (*n* = 5). Representative data are from one of three experiments with similar results. The gated human CD45-positive and mouse CD45.1-negative viable cells were analyzed. Isotype-matched IgGs served as controls for flow cytometry. (**F**) Myeloid differentiation of miCD34^+^ HSCs after transplantation into NSG mice at 12 weeks (top) and 16 weeks (bottom, *n* = 3).

**Figure 5 ijms-21-04249-f005:**
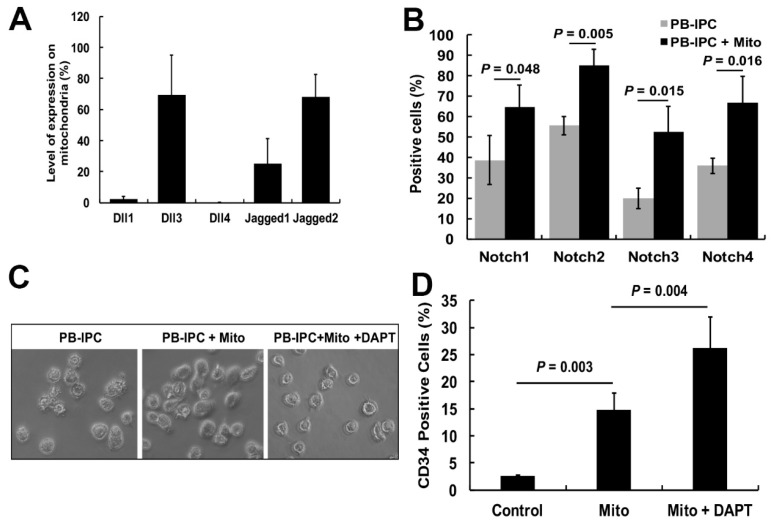
Notch pathway in the differentiation of PB-IPCs toward miCD34^+^ HSCs. (**A**) Analysis of Notch ligands on platelet-derived mitochondria by flow cytometry (*n* = 3). (**B**) Expression of Notch receptors on PB-IPCs by flow cytometry. PB-IPCs were treated with mitochondria for 7 days and collected for flow cytometry. Untreated PB-IPCs served as control. *n* = 3. (**C**) Phase-contrast images show the morphology of PB-IPCs (left) and treated PB-IPCs in the presence of mitochondria (middle) and mitochondria + DAPT (right). Original magnification: ×200. (**D**) CD34 expression upregulation after the treatment with mitochondria and/or DAPT. DAPT-treated PB-IPCs served as control. *n* = 4. Data represent mean ± SD.

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
