# Peer review of "Generation of Hematopoietic-Like Stem Cells from Adult Human Peripheral Blood Following Treatment with Platelet-Derived Mitochondria"

_ijms, 2020, doi:10.3390/ijms21124249_

Round 1

Reviewer 1 Report

In their manuscript the authors describe the generation of hematopoietic-like stem cells starting from human peripheral blood cells treated with platelet-derived mitochondria. The study is interesting and of potential clinical relevance, however the following points need to be addressed.

Major points

1) In the result section, almost all the representative flow cytometry images are reported without adding a corresponding plot of the mean values and statistical analyses. This information must be added. In addition, considering a general audience, the data could result more clear if presented in form of histogram.  For the sake of clarity, I also suggest to specify which is the meaning of positivity or negativity for the selected markers; for example what does it means that “CD34+ (miCD34+) cells displayed a phenotype of CD34+CD38-/lowCD45RA-CD49f+CD90+Flt3-/lowCD7+CD10+CD71+BAH1-/low “? Other examples of this kind are found at lines 237 and 239. It is difficult to grasp the take at home message if the reader is not expert in the field.

2) It is unclear how many biological replicates and technical replicates have been performed in each experiments. The indications given in the captions are somewhat contradictory: for example in fig 2g the authors indicate “n=4” and also state “Data are presented as mean ± SD from three experiments”.

4) As presented, the method section is somewhat confounding. The procedure for platelet isolation is not reported. The procedure for isolation of miCD34+ cells is described in lines 266-268 of the discussion section and should be moved in the method section. The list of the conjugated antibodies and other fluorescent probes used in the various staining procedures is repeated in par 4.3 and par 4.4. I suggest to separate the list of fluorescent probes from the description of the experimental procedures. Also, to improve the readability of the text and the results, I suggest better indicating what the various markers are used for. As I recommended at point 1, please consider a general readership and not only experts of flow cytometry or hematology. In addition, in lines 454-56 the authors mention how they have selected and analyzed the cells from human origin in the samples of mouse as follows: “The gated human leukocyte common antigen CD45 positive and mouse CD45.1-negative viable cells were analyzed for characterization with different lineage-specific surface markers…”, this sentence is quite ambiguous and unclear, please fix it.

6) In the statistical analyses section the authors mention the parametric student T test, have the authors assesse3d if the data distribution satisfies the conditions for this type of test?

5) The authors discuss but do not report any mechanistic insight. Since in lines 278-281 the authors mention some unpublished data on Notch receptor 1 signaling as a putative mechanism, I suggest deepening this aspect and adding the results to the present manuscript.

Minor points

1) English language and syntax need to be improved, here some examples:

Line 16-17, “…novel function of mitochondria directly contributed to cellular reprogramming.” It is unclear, maybe the authors mean: “novel function of mitochondria directly contributing to cellular reprogramming.”

Line 50, “we developed the Stem Cell Educator (SCE) therapy, which circulates a patient’s blood through a blood cell separator, co-cultures the patient’s immune cells with adherent CB-SCs in vitro, and returns “educated” immune cells to the patient’s circulation.” It is unclear, maybe the authors mean: “we developed the Stem Cell Educator (SCE) therapy, which consist in isolating patient’s blood cells through a blood cell separator, co-culturing the patient’s immune cells with adherent CB-SCs in vitro, and returning the “educated” immune cells to the patient’s circulation.”

Line 88, “Using an optimized panel of cell markers [21], mitochondrion-induced CD34+ (miCD34+) cells displayed a…” It is unclear, maybe the authors mean: “Using an optimized panel of cell markers [21], we found that mitochondrion-induced CD34+ (miCD34+) cells displayed a….”

Line 345, “PB-IPC’s nuclear were washed” it is incorrect, please amend.

Author Response

Dear Reviewer,

We appreciate your kind consideration and constructive comments! Please see our responses in an attached document. 

Best regards,

Yong

Reviewer 2 Report

  1. The plagiarism is 35%, it should be decreased.
  2. The sample used in the manuscript was human blood. Please provide approved protocol information.
  3. The age range was from 18 to 72 years old, it’s known that aging induced many organ mitochondrial dysfunctions including platelet mitochondria. Are the bloods pooled together to isolate PBMC? If yes, how do you think the influences of aging on generation of hematopoietic-like stem cells? If not, it’s stronger if you show your data separately?
  4. The blood used to isolate platelet mitochondria was also purchased from New York Blood Center but not giving detail information such as age. Mitochondria is very sensitive to aging, so does author consider this concern?
  5. Line 345, “PB-IPC’s nuclear were washed with PBS at 500g for 5 mins at 4 °C.” no data to be related nuclear and no method to isolate nuclear?
  6. Cell surface antigen staining is different from intracellular staining, please clarify your methods.
  7. Regarding statistics, what software you used and why you performed by the two-tailed paired student’s t-test? Did you check the normality in same group?
  8. It’s common that cytochrome c is a marker for checking mitochondrial damage, question for authors that why you didn’t still use gated cells by using Mito Tracker to test mitochondrial purification?
  9. May readers know what is the potential mechanisms by which platelet-derived mitochondrial treatment resulted in differentiation into CD4 T cell only?
  10. Actually, I have been thinking about what and how mitochondria stimulates PB-IPC to be differentiation into CD4 T cell? If you used mitochondrial lysate same protein amount as used mitochondria to treat PB-IPC, what happening?
  11. I have last concern about isolated mitochondria. What is the difference between platelet-derived mitochondria and other blood cells-derived mitochondria or isolated mitochondria from organs like heart, liver, skeletal muscle and so on?

Author Response

(The authors gave the same response as above.)

Round 2

Reviewer 1 Report

Although the manuscript has been improved some clarifications are still needed.

Lines 85-91: It Is unclear why the authors report in the result section the findings of a previous work on the use of platelet derived mithochondria for improving the function of human pancreatic cells. Please clarify.

Line 95: please describe the ratio of using the panel of cell markers. What kind of information can we get from It?

Line 96: what does It means that the cd34+ cells exhibit those markers?

The results shown in the new fig. 5 should be discussed in more detail. What Is the rational of using the notch inhibitor? Why notch inhibition increase the percentuale of cd34+ cells? The authors should also explicit the proposed mechanistic model via mithochondria (notch ligand)/PB-ipc cells (receptor) interaction.

Author Response

Dear Reviewer,

We sincerely appreciate your kind consideration and constructive comments for us to improve the quality of this manuscript! Please see our responses in the attachment. 

Best regards,

Yong

Reviewer 2 Report

Thanks for your nice revised version ! I am happy that you answered all of my concerns carefully and clearly! However, I still have one concern regarding human blood samples. Please tell us the reason why you selected the age range from 18 to 72 years old? Based on the information from the new version, author has right to decide what age range will be limitation. In addition, the answer regarding my concern 3, "All buffy coats and apheresis platelets were utilized at the individual setting for each experiment (not pooled)" means that each sample has separately experimental parameters. If so and not pooled, does author analyze the column statistics to determine if the data are normally distributed? Or may author provide the statistic data like age range 18-35 years old, 36-60 years old and 61-72 years old?

Author Response

(The authors gave the same response as above.)

Round 3

Reviewer 1 Report

The authors have satisfactorily responded to all my comments. 

Reviewer 2 Report

Great revised version and it is ready to be accepted!

but one concern is that your revised version and your supplemental materials had only one figure (S1), however, you should have 4 figures based on your cover letter. Please provide full figures in your final supplement material!